

# A technique for volumetric incoherent scatter radar analysis

Johann Stamm[1], Juha Vierinen[1], Björn Gustavsson[1], and Andres Spicher[1]

[1]Institute for physics and technology, University of Tromsø, Tromsø, Norway

**Correspondence:** Johann Stamm (johann.i.stamm@uit.no)

**Abstract.** Volumetric measurements of the ionosphere are important for investigating spatial variations of ionospheric features, like auroral arcs and energy deposition in the ionosphere. In addition, such measurements make it possible to distinguish between variations in space and time. While spatial variations in scalar quantities such as electron density or temperature have been investigated with ISR before, spatial variation in the ion velocity, which is a vector quantity, has been hard to measure. The upcoming EISCAT3D radar will be able to do volumetric measurements of ion velocity regularly for the first time. In this article, we present a technique for relating volumetric measurements of ion velocity to neutral wind and electric field. To regularize the estimates, we use Maxwell's equations and fluid-dynamic constraints. The study shows that accurate volumetric estimates of electric field can be achieved. Electric fields can be resolved at altitudes above 120 km which is the altitude range where auroral current closure occurs. Neutral wind can be resolved at altitudes below 120 km.

## 1 Introduction

It would be of huge importance to measure the how electric fields in and around auroral arcs vary in time and space. This would allow us to gain new knowledge on the evolution of currents in Cowling-channels, the closure of Birkeland currents and ultimately the dynamics of magnetosphere-ionosphere coupling in the auroral regions. To investigate the spatial variation of the ionospheric electrical fields and currents, it is necessary to measure how physical quantities vary over a volume in the ionosphere (e.g. McCrea et al., 2015).

Investigating the spatial variation of the ionosphere can be done in two different ways: multi-beam scanning or aperture synthesis radar imaging (ASRI). With multi-beam scanning/volumetric imaging (Semeter et al., 2009; Nicolls et al., 2014; Swoboda et al., 2014, 2017), the radar beam is pointed in different directions to measure the local states in the ionosphere. Multi-beam scanning covers a large region in the ionosphere, and is thereby useful for investigating large-scale structures. With ASRI, the phase difference in received signal between receivers is used to investigate small-scale structures inside of the radar beam (see e.g. Hysell and Chau, 2012). In this paper, we investigate the multi-beam scanning with E3D. For ASRI with E3D, we refer to Stamm et al. (2021b).

A phased array is an array of (dipole) antennas where the beam can be steered by changing the phase of the transmitted or received signals. Combined with electronic control of the phases at every antenna, the beam steering can be performed between two consecutive pulses (e.g. Wirth, 2001). The AMISR radars (Valentic et al., 2013; Heinselman and Nicolls, 2008) were the first ISRs that combined these two, making it possible to perform measurements of scalar ionospheric parameters,



such as electron density $n_e$, electron temperature $T_e$ and ion temperature $T_i$ in some tens of seconds (Semeter et al., 2009). By assuming that the electric field along magnetic field lines is constant and that the field-aligned ion flow is completely constant, the variation in Doppler shift can be used to estimate horizontal variations in electric field (Nicolls et al., 2014). However, full

volumetric measurements of vector parameters require multiple receivers. At least one receiver for every component of the ion velocity vector is needed. This will be able with E3D when it is finished (McCrea et al., 2015).

With the first three sites of E3D, volumetric measurements of ion velocity will become possible. The core site with combined transmitter and receiver is going to be in Skibotn, Norway, and two remote receiver sites are built in Kaaresuvanto, Finland and Kaiseniemi, Sweden. Each site will have a phased array, which will be built with up to 109 hexagonal subarrays consisting of

91 crossed dipole antennas each. In Skibotn, additional 10 outrigger subarrays are built for interferometry (Kero et al., 2019).

The technique for estimating electric field and neutral wind from ion velocity has been based on determining the electric field at high altitudes where the ion drift is dominated by ExB drift. Then, the electric field has been assumed to be constant along the magnetic field line so the neutral wind could be estimated at lower altitudes. This technique was introduced by Brekke et al. (1973) and has been used in many studies of the neutral wind (Brekke et al., 1974, 1994; Brekke, 2013; Heinselman and

Nicolls, 2008; Nygrén et al., 2011, 2012). However, for analyzing a vector field, the method has to be adjusted because only one beam will be field-aligned.

In this work we present a technique to estimate the three-dimensional variation of electric fields and neutral winds from multi-static ISR measurements of ion velocities. A volumetric model makes it possible to use Maxwell's equations and the continuity equation for the neutral wind to constrain the estimates. The work is a three-dimensional generalization of the

work of Stamm et al. (2021a) that investigated the possibility of using an field-aligned profile with E3D measurements of ion velocity to find estimates of electric field and neutral wind. When generalizing, one has to take into account that most of the measurements are not aligned with the magnetic field. With the improvements of Heinselman and Nicolls (2008); Nygrén et al. (2011); Stamm et al. (2021a), we will develop a model which can be used to analyze the three-dimensional vector fields of neutral wind and electric field.

The paper is organized as follows: The general technique to obtain neutral wind and electric field from ion velocity measurements is described in Sect. 2. The framework for volumetric measurements and estimates is described in Sect. 3. Our chosen setup of the measurements and discretization of the neutral wind and electric field estimates is shown in Sect 4. Section 4.1 discusses the uncertainties in the measurements, applicability of the assumptions and uncertainties of the estimates. A simulation of ion drift measurements is given in Sect. 5 , followed by a discussion in Sect. 6.

## 2 Velocity of ions and neutrals and electric field

The estimation of neutral wind and electric field consists of three steps: First measuring Doppler shifts, then finding the ion velocity vectors, and finally estimating neutral wind and electric field.

Incoherent scatter radar measurements are performed by transmitting a powerful radio wave and measuring the spectrum of the scattered signal, which at frequencies much larger than the plasma frequency contains information about the plasma that





scatters the radio waves. Due to collective motion of the ions, the spectra are Doppler shifted. This shift is used to obtain the
ion velocity component parallel to the Bragg scattering vector $\boldsymbol{k}_{\mathrm{B}}$ which is equal to the difference between wave vectors of the
scattered and transmitted wave (see also Beynon and Williams, 1978). Figure 1 illustrates the characteristic geometry of E3D
together with the wave vectors that the ion velocity is measured along.

The relationship between a measurement of the Doppler shift $w$ and the ion velocity vector for transmitter-receiver pair $p$ is

$$w_p = \frac{\boldsymbol{k}_p}{|\boldsymbol{k}_p|} \cdot \boldsymbol{v}. \tag{1}$$

A set of Doppler shift measurements $\boldsymbol{w}^\top = [w_1, ... w_P]$ of the same volume from $P$ pairs can be combined to system

$$\boldsymbol{w} = \mathbf{K}\boldsymbol{v} + \boldsymbol{\xi_w}, \tag{2}$$

where $\mathbf{K}^\top = [\boldsymbol{k_1}/|\boldsymbol{k_1}|, ..., \boldsymbol{k_P}/|\boldsymbol{k_P}|]$ is the theory matrix, and $\boldsymbol{\xi_w}$ is a vector containing the noise terms. If the measurements
are sufficiently linearly independent, the ion velocity can be found with the method of least squares (cf. Aster et al., 2013;
Risbeth and Williams, 1985).

Ion velocity is determined by the ion momentum equation. At ionospheric altitudes, the dominant terms are Lorentz force and
collision with neutrals while the terms for advection, gravity and pressure gradients are negligible. When assuming steady-state
conditions, the ion momentum equation can be written as

$$0 = q_{\mathrm{e}} n_{\mathrm{e}} \left( \boldsymbol{E} + \boldsymbol{v} \times \boldsymbol{B} \right) - n_{\mathrm{e}} m_{\mathrm{i}} \nu_{\mathrm{in}} \left( \boldsymbol{v} - \boldsymbol{u} \right), \tag{3}$$

where $q_e$ is the unit charge, $n_e$ is the electron density, $\boldsymbol{E}$ is the electric field vector, $\boldsymbol{B}$ is the magnetic field, $m_i$ is the average
mass of ions, $\nu_{\mathrm{in}}$ is the momentum transfer collision frequency between ions and neutrals, and $u$ is the neutral wind velocity.

To simplify the algebra, we rewrite the cross product with a matrix multiplication. We introduce the matrix

$$\mathbf{B}_g = \begin{bmatrix} 0 & B_z & -B_y \\ -B_z & 0 & B_x \\ B_y & -B_x & 0 \end{bmatrix}, \tag{4}$$

where $x, y$ and $z$ are the axes of the geographic coordinate system, that are east, north and up. This allows us to rewrite the
cross product as $\boldsymbol{v} \times \boldsymbol{B} = \mathbf{B}_g \boldsymbol{v}_g$ where the subscript $g$ shows that the matrix and vector are in geographic coordinates.Now, the
momentum equation can be rewritten as

$$\left( \mathbf{I} - \frac{\kappa}{B} \mathbf{B}_g \right) \boldsymbol{v}_g = \frac{\kappa}{B} \boldsymbol{E}_g + \boldsymbol{u}_g, \tag{5}$$

where

$$\kappa = \frac{q_{\mathrm{e}} B}{m_{\mathrm{i}} \nu_{\mathrm{in}}}. \tag{6}$$

is the ion mobility and $\mathbf{I}$ is the identity matrix. Inverting the matrix on the left side is simplified by transforming into local
magnetic coordinates perpendicular to the magnetic field towards east and antiparallel. The third component completes the





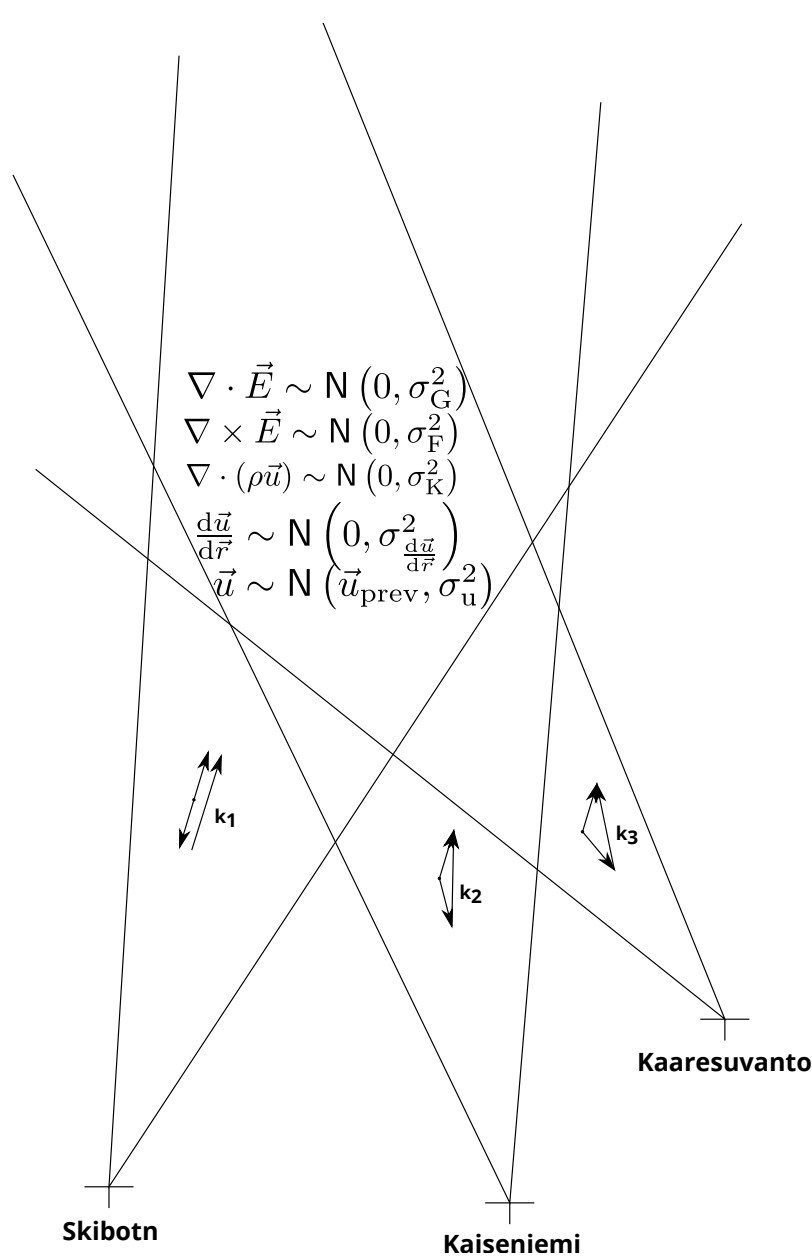

**Figure 1.** The figure shows geometry and assumptions on E3D volumetric measurements. The figure is not to scale or angle.





right handed system and will be referred to as northward. The transformation matrix from local geomagnetic to geographic coordinates is

$$
\mathbf{R} = \begin{bmatrix} \cos\delta & \sin I \sin\delta & -\cos I \sin\delta \\ -\sin\delta & \sin I \cos\delta & \cos I \cos\delta \\ 0 & \cos I & \sin I \end{bmatrix} \tag{7}
$$

for declination $\delta$ and magnetic dip angle $I$ (Heinselman and Nicolls, 2008). The matrix $\mathbf{R}$ is a rotation matrix, which means that $\mathbf{R}^{-1} = \mathbf{R}^{\top}$. The matrix on the left hand side of Eq. (5) can then be written as $\mathbf{R}\mathbf{C}_m^{-1}\mathbf{R}^{\top}$, where

$$
\mathbf{C}_m = \frac{1}{1+\kappa^2} \begin{bmatrix} 1 & -\kappa & 0 \\ \kappa & 1 & 0 \\ 0 & 0 & 1+\kappa^2 \end{bmatrix}. \tag{8}
$$

The momentum equation can now be written as

$$
\boldsymbol{v}_g = \mathbf{R}^{\top}\mathbf{C}_m\mathbf{R}\boldsymbol{u}_g + \frac{\kappa}{B}\mathbf{R}^{\top}\mathbf{C}_m\boldsymbol{E}_m, \tag{9}
$$

indicating that we will estimate the electric field in local magnetic coordinates.

## 3   Vector field estimation model and grid

This section defines the model that will be used to estimate electric field and neutral wind from multi-beam multistatic ISR observations of ion velocity. The electric field and neutral wind have three components each which have to be found from a discrete set of three components of ion wind. This gives six unknowns for three measurements. In addition to relating the ion

velocity with the electric field and neutral wind, therefore also constraints are applied to find a more stable solution.

The discretization of the problem should keep most of its important features. The volume unknown is represented by discrete basis functions where we use a discretization corresponding to boxcars (voxels) in a desired coordinate system. This simplifies the search for discretization to find one coordinate system for each unknown. It is an advantage for computation speed to let the discretization be as coarse as possible because fewer parameters have to be estimated.

The electric field is strongly affected by the electric conductivities. This means that the fields are stronger in directions where the conductivity is low. Since the conductivity is much higher along the magnetic field than perpendicular to it (Brekke, 2013), electric fields and their variations are expected to mainly be in the perpendicular direction for higher altitudes. To avoid aliasing-type problems it is preferable to use a discretization that is aligned with the magnetic field.

The neutral wind is expected to vary predominantly perpendicular to gravity and therefore following the surface of Earth. A

geographic oriented coordinate system is therefore an advantage for the neutral wind.

This means that the preferred coordinate systems for the discretization of electric field and neutral wind are different. We introduce now the discretization. We start with the measurements of the ion velocity. Here, for measurement $\ell$, the measured





ion velocity $\boldsymbol{v}_\ell$ is considered as an integral over the probed volume indicated with the function $\beta_\ell$. The measurement can be written as

$$\boldsymbol{v}_\ell = \iiint_V \boldsymbol{v}(\boldsymbol{r})\beta_\ell(\boldsymbol{r})\,\mathrm{d}V + \boldsymbol{\xi_\ell}, \tag{10}$$

where $\boldsymbol{\xi}_\ell$ is a vector which contains the errors of the ion velocity vector measurements, that are the errors of the solution of Eq. (2). Equation (10) can be expanded using the momentum equation, Eq. (9). This gives

$$\boldsymbol{v}_\ell = \iiint_V \mathbf{R}^\top \mathbf{C}_m \mathbf{R}\boldsymbol{u}_g(\boldsymbol{r})\,|\det\mathbf{J}_u|\,\beta_\ell(\boldsymbol{r})\,\mathrm{d}V + \iiint_V \frac{\kappa}{B}\mathbf{R}^\top\mathbf{C}_m\boldsymbol{E}_m\,|\det\mathbf{J}_E|\,\beta_\ell(\boldsymbol{r})\,\mathrm{d}V + \boldsymbol{\xi_\ell}, \tag{11}$$

where $\mathbf{J}$ is the Jacobian from the coordinate system of the ion velocity to that one indicated by the subscript, $u$ for neutral wind and $E$ for electric field. Then, the unknown continuous vector fields are discretized by replacing them with sums of basis functions $\boldsymbol{\Phi}_j$ and $\boldsymbol{\Psi}_j$:

$$\boldsymbol{E} \approx \sum_{j=1}^{N_E} \eta_j \boldsymbol{\Phi_j} \tag{12}$$

and

$$\boldsymbol{u} \approx \sum_{j=1}^{N_u} \Gamma_j \boldsymbol{\Psi_j}. \tag{13}$$

This converts the continuous vectorfield to a discrete form where the coefficients $\eta_j$ and $\Gamma_j$ are our new set of unknowns. They are constant over the integrated volume and can therefore be taken out of the integral. We will now define the variables

$$a_{\ell j}^E = \iiint_V \frac{\kappa}{B}\mathbf{R}^\top\mathbf{C}_m\boldsymbol{\Phi}_j\,|\det\mathbf{J}_E|\,\beta_\ell(\boldsymbol{r})\,\mathrm{d}\boldsymbol{r} \tag{14}$$

and

$$a_{\ell j}^u = \iiint_V \mathbf{R}^\top\mathbf{C}_m\mathbf{R}\boldsymbol{\Psi}_j\,|\det\mathbf{J}_u|\,\beta_\ell(\boldsymbol{r})\,\mathrm{d}\boldsymbol{r}. \tag{15}$$

Equations (14) and (15) let us write Eq. (11) as

$$\boldsymbol{v}_\ell = \sum_{j=1}^{N_u} a_{\ell j}^u \Gamma_j + \sum_{j=1}^{N_E} a_{\ell j}^E \eta_j + \boldsymbol{\xi}_\ell \tag{16}$$

which can be recognized as a matrix equation $\boldsymbol{v} = \mathbf{A}_E\boldsymbol{\eta} + \mathbf{A}_u\boldsymbol{\Gamma} + \boldsymbol{\xi}$. If we define the unknowns as one single vector $\boldsymbol{x}^\top = \left[\boldsymbol{\Gamma}^\top, \boldsymbol{\eta}^\top\right]$ and stack the matrices $\mathbf{A}^\top = \left[\mathbf{A}_u^\top, \mathbf{A}_E^\top\right]$, the equation relating the measurements to the unknowns becomes

$$\boldsymbol{v} = \mathbf{A}\boldsymbol{x} + \boldsymbol{\xi}. \tag{17}$$

The equation, can be recognized as a standard linear inverse problem, and is what we develop a general physics-based solution to in this paper.





The nature of the problem is underdetermined as shown by the earlier works (e.g. Brekke et al., 1973; Semeter et al., 2009; Nygrén et al., 2011, 2012; Nicolls et al., 2014; Swoboda et al., 2017; Stamm et al., 2021a). We therefore have to use regularization. Here, we will show that for the electric field and neutral wind we can use fundamental physical law to obtain regularization terms similar to Tikhonov regularization. This both gives a less noisy solution and a forces it to be physically reasonable.

By using Gauss' law $\nabla \cdot \boldsymbol{E} = 0$ for a charge-neutral plasma and Faraday's law $\nabla \times \boldsymbol{E} = \boldsymbol{0}$ for a time-stationary magnetic field, we are adding 4 equations for every unknown vector of the electric field.

For the neutral wind, we use the continuity equation $\nabla \cdot (\rho \boldsymbol{u}) = 0$, where $\rho$ is the mass density of neutral particles. Also, we assume that the acceleration of the neutral wind is small. This means that when the same particles have moved for some time, and thereby distance, they have the same velocity. Further on, this implies that the spatial variation of the neutral wind vector field is small. We implement this approximation by assuming that the first order differences of the neutral wind components in all directions are smaller than some parameter $1/\alpha$. These constraints are mathematically equivalent to first order Tikhonov regularization (Aster et al., 2013; Roininen et al., 2011).

With small neutral wind accelerations, one can also argue to use previous estimates of the neutral wind as prior assumption of the next estimate of neutral wind. This corresponds to a zeroth order Tikhonov regularization and would then be similar to a Kalman filter, or to the approach introduced by Nygrén et al. (2011).

Many of the regularization termswe introduce contain spatial derivatives in multiple dimensions at the same time. For example, each component of Faraday's law uses derivatives in two directions, as illustrated in Fig. 2. Since these derivatives in this case are not symmetrical, we use a weighting of the derivatives in both directions. They are approximated by

$$\frac{\mathrm{d}E_x}{\mathrm{d}y}(y) \approx W_1 \frac{E_x(y + \Delta y_1) - E_x(y)}{\Delta y_1} + W_2 \frac{E_x(y) - E_x(y - \Delta y_2)}{\Delta y_2}. \tag{18}$$

for the example of electric field in x-direction. In the equation, $W_1$ and $W_2$ are weights. We note that the separation in the grid is varying because the grid may be curved and stretched. Therefore we have to take into account that $\Delta y_1 \neq \Delta y_2$.

Additionally, when differentiating in different dimensions, there appear border issues since in some cases the derivatives can only be found in some directions, see Fig. 2. Mathematically, the solutions to this problem differ in which weights $W_1$ and $W_2$ are used. We are aware of three possible solutions. The first is to ignore the derivatives passing the border. Then, one of the weights is zero, which is shown as the blue line in Fig. 2.

Another possibility is to take the border-passing derivatives as stochastic variables, that is that e.g.

$$\frac{E_x(y) - E_x(y - \Delta y_2)}{\Delta y_2} \sim N\left(0, \sigma_{\Delta E}^2\right). \tag{19}$$

A third possibility is to weigth the two derivatives in another way, for example by focusing on those inside of the borders. An example is illustrated by the cyan arrows in Fig. 2.

The problems described above do not apply to the one-dimensional derivatives in the first order Tikhonov regularization for the neutral wind. In this case we simply use the definition of the derivative.



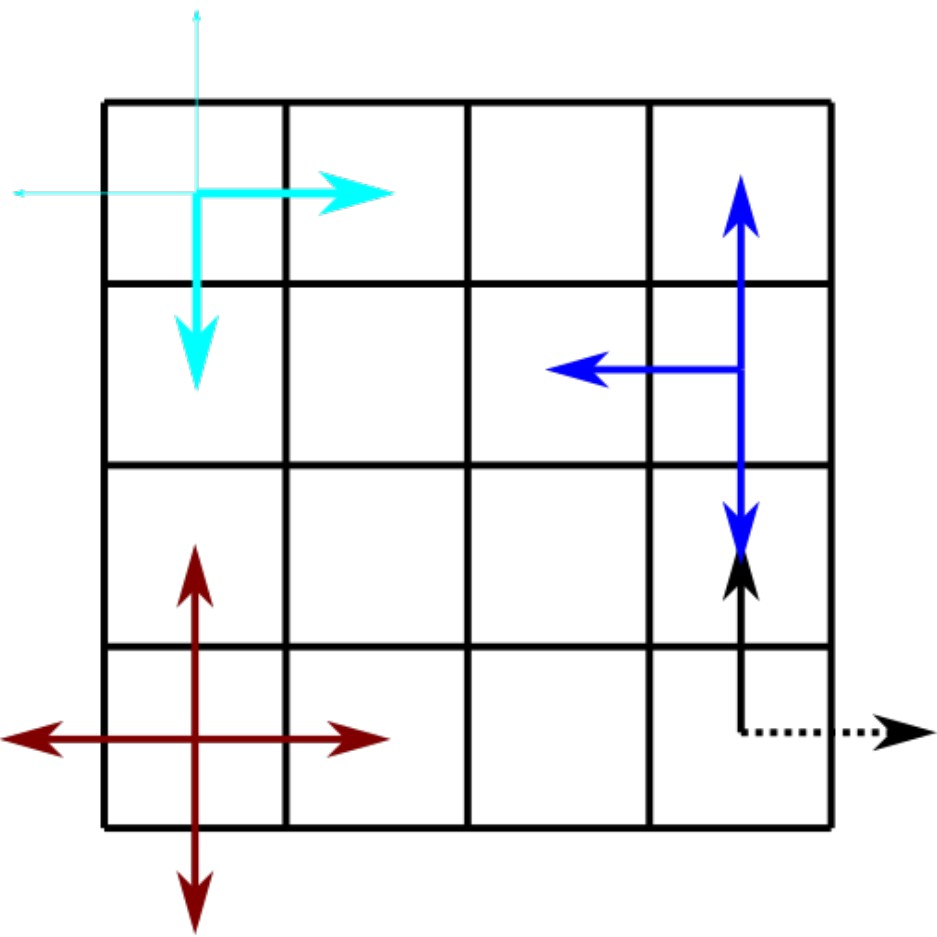

**Figure 2.** Problems that arise at the borders of the grid. When using the definition of the derivative, at the one side, the derivative over the border can not be included directly (black arrows). Possible solutions to the border problem for symmetric derivatives are also shown in the figure (cyan, blue, brown arrows).

These regularizing constraints add several terms to our inverse problem. The physics-based regularized function we are minimizing is

$$(\boldsymbol{m} - \mathbf{A}\boldsymbol{x})^\top \Sigma_m^{-1} (\boldsymbol{m} - \mathbf{A}\boldsymbol{x}) + (\nabla \times \boldsymbol{E})^\top \Sigma_F^{-1} (\nabla \times \boldsymbol{E}) + (\nabla \cdot \boldsymbol{E})^\top \Sigma_G^{-1} (\nabla \cdot \boldsymbol{E}) + (\nabla \cdot (\rho \boldsymbol{u}))^\top \Sigma_K^{-1} (\nabla \cdot (\rho \boldsymbol{u}))$$

$$+ \left(\frac{\mathrm{d}\boldsymbol{u}}{\mathrm{d}\boldsymbol{r}}\right)^\top \Sigma_{\frac{\mathrm{d}\boldsymbol{u}}{\mathrm{d}\boldsymbol{r}}}^{-1} \left(\frac{\mathrm{d}\boldsymbol{u}}{\mathrm{d}\boldsymbol{r}}\right) + (\boldsymbol{u} - \boldsymbol{u}_{\mathrm{prev}})^\top \Sigma_{\frac{\mathrm{d}\boldsymbol{u}}{\mathrm{d}t}}^{-1} (\boldsymbol{u} - \boldsymbol{u}_{\mathrm{prev}}). \quad (20)$$

Here, the covariance matrices in the different regularization terms fill the same role as the regularization parameter in a standard Tikhonov regularization.They balance how tightly the solution fits the constraints relative to how well they fit the observations.



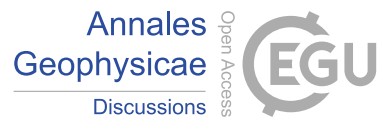

It is possible to rewrite this on matrix form as

$$\boldsymbol{v}_{\mathrm{R}} = \mathbf{A}_{\mathrm{R}}\boldsymbol{x} + \boldsymbol{\xi}_{\mathrm{R}}. \tag{21}$$

where the extended theory-matrix is $\mathbf{A}_{\mathrm{R}}^{\top} = \left[\mathbf{A}^{\top}, \mathbf{L}^{\top}\right]$. Here, the matrix $\mathbf{L}$ is the regularization matrix which contains all the regularization terms constraining the problem.

## 4 Model simulation

To analyze the resolution and accuracy the proposed estimation technique provides, we perform a simulation of the system. Here we use different grids for ion drifts, electric field and neutral wind.

For the simulated measurements, we use an experiment consisting of 5x7 beams, as illustrated in Figs. 3 and 4. The beams are pointed evenly as a fan with zenith angles from 13° southward (approx. magnetic field-aligned), with a spacing of 3° to 5° northward, and every 2.5° between 5° westward and 5° eastward. In every beam, we measure with ranges every five km range resolution from 90 to 210 km range.

We model the measurements using Gaussian beam-pattern perpendicular to the range direction and triangular weights along range. The vertices of the triangle are placed in the center of the next range gate. At the nearest and furthest range, the triangles are symmetric. The Gaussian functions are centered around the line of sight with an standard deviation of 1° corresponding to the HPBW. The Gaussian is truncated at 2 standard deviations and normalized such that it still integrates to 1.

The grid for the neutral wind uses geographic coordinates, as shown in Figs. 3 and 4. The grid centers are placed every 0.15° between 68.9° and 69.5° latitude and every 0.3° between 19.8° and 20.7° longitude. In altitude, we place the centers every tenth kilometer between 90 and 210 km.

For the electric field, we choose a special coordinate system. One axis is field-aligned and therefore slightly curved, as the magnetic field is not completely straight. However in a short height range, as in Figs. 3 and 4, the curvature is not visible. The other axes consist of geographic latitude and longitude at the surface of Earth. We place the horizontal grid centers for the electric field every 0.1° within 69.3°-69.9° in latitude and every 0.2° within 20.0° and 21.0° in longitude on the surface of Earth. The grid contains 7 voxels in latitude and 6 voxels in longitude. Along the magnetic field axis, the centers are placed every tenth km between 90 and 210 km.

### 4.1 Uncertainties in ion velocity vectors

In this section we will calculate the uncertainty in estimates of electric field and neutral wind for the example setup outlined in Sect. 4. In order to find the accuracy of the solution, we must first estimate the uncertainty in the measurements, that is in both observations and constraints. The accuracy of ion-drift observations is well understood, but depend on the ionospheric conditions, primarily the electron density. Thus, the uncertainty varies over time, space and with the component considered (e.g. Stamm et al., 2021a). Some assumptions are therefore necessary. Here, we performed similar calculations as Stamm et al. (2021a), but using parameters of E3D when the full first stage is finished, that is a HPBW of 1°, transmit power of 5 MW, and

**Figure 3.** Longitude-height-view of experimental layout. The radar beams are shown in blue, the grid for neutral wind in black and the grid for electric field in red.





**Figure 4.** Latitude-height-view of experimental layout.





transmit/receive gains of 43 dB. We also increased the averaging in range of the measurements to 4500 m in order to fit better
to the setup in this study. With an integration time of 2 s, the horizontal ion drift can be measured with around 20 m/s accuracy
in horizontal and 5 m/s in vertical direction. This makes a full loop over all 35 beams take 70 seconds.

When we calculate the uncertainties, we have neglected the effects of cases where transmit and receive beam only overlap
partially, decreased transmit/receive gains for tilted beams and scattering angles below 90°. All these effects will increase the
uncertainty in ion drift observations, but not significantly.

### 4.2 Regularization parameters

The next step is to select suitable weights for the regularization terms, that are Maxwell's laws, the continuity equation and the
assumption of low neutral wind acceleration. This can be interpreted as estimating the uncertainty in uncovered terms or the
additional constraints they impose. The equations for Gauss' law are equivalent to saying that the expected ionospheric charge
density is zero with a variance that corresponds to some value of $\rho/\varepsilon_0$, where $\rho$ is the net charge density and $\varepsilon_0$ is the permittivity
in vacuum. The uncertainty in the Gauss' law regularization is thereby decided by the amount of plasma charge-neutrality. We
can, for example, assume that the usual deviation from charge neutrality is 1 to a million, meaning that for $10^6$ electrons one
is missing a positive charge. If the electron density is $10^{11} \mathrm{m}^{-3}$, around $10^5$ electrons do not have a corresponding positive
charge. Then, the net charge in the plasma is on the size of $10^{-14}$ C/m³. In sum, we assume that $\nabla \cdot \boldsymbol{E} \sim \mathcal{N}(0, (10^{-3}\mathrm{V/m}^2)^2)$.

In Faraday's law, the uncovered term is the time derivative of the magnetic field. In general, time variations in the magnetic
field are mostly quite slow, but sometimes it changes very rapidly, for instance during substorms. To include also these condi-
tions, we will use a rapid changing magnetic field as a measure. As an example, we use ground-based magnetometer data for
interplanetary shock in 2012 as shown by Belakhovsky et al. (2017). Of the shown magnetometer measurements, the strongest
change in the magnetic field was measured in Ivalo. There, in one minute, the x-component of the magnetic field increased
by 600 nT, giving an increase of 10 nT/s. Through testing, even this rapid change seems too small. We will assume that the
time-derivative of any magnetic field component is distributed as $\frac{\mathrm{d}B_{(x,y,z)}}{\mathrm{d}t} \sim \mathcal{N}(0, (300\mathrm{nT/s})^2)$.

The continuity equation for neutrals is

$$\frac{\mathrm{d}\rho}{\mathrm{d}t} + \nabla \cdot (\rho \boldsymbol{u}) = 0 \tag{22}$$

We assume that the strongest changes in neutral density are caused by gravity waves which in turn affect the electrons in a
similar manner, one can use changes in electron density to obtain information about the change in neutral density. When doing
this, it is very important to be cautious of changes that would only affect the electron density. Therefore it will be advantageous
to estimate the slope of neutral density over a short time period and in geomagnetically quiet conditions. One example of this
was measured with EISCAT UHF in the of 10. September 2005. There, in the F region, the electron density varied up to 50
% in around 10 minutes (Nygrén et al., 2015). Transferring this to neutral density at 100 km altitude, this would correspond
to a change in the order of 50 %, which is about 0.6 kg/m³, provided that the assumptions hold. We therefore assume that
$\frac{\mathrm{d}\rho}{\mathrm{d}t} \sim \mathcal{N}(0, (10^{-3}\mathrm{kg/m}^3\mathrm{s})^2)$. This corresponds to about $10^{22}$ molecules of molecular oxygen per cubic meter and seconds.





In sum, with these variances, we assume that in 67% of the time, the net charge density in the plasma volume is lower than $10^{-14}$ C/m³, the magnetic field varies less than 300 nT/s and the neutral mass density varies less than 1 g/(m³s).

In addition, we need to have some estimate for the cases where we consider the derivative of electric field or neutral wind over the borders of our grid and for the constraint of small neutral wind accelerations. We implement both of these in the same way where we let the gradient be a stochastic variable with a variance as in Eq. (19). For the electric field, we use the uncertainties

that Stamm et al. (2021a) used in the field-aligned one-dimensional case and extend the use to all three dimensions. This corresponds to assuming that the standard deviation of the electric field in the corresponding cases is smaller than 20 mV/m per 2500 m.

For the variance of the neutral wind gradients we use approximate variations in measurements taken with a scanning Doppler imager as shown by Zou et al. (2021). Here, it appears that the latitudinal variation in the horizontal neutral wind components

is mostly below 100 m/s per degree latitude, corresponding to about 2 m/s per 10 km. We tighten this constraint to 1 m/s per km. In vertical direction we use a looser constraint of 20 m/s per km to allow for wind shear. This constraint of the neutral wind is applied to the whole volume and corresponds directly to first order Tikhonov regularization.

In addition, we constraint the magnitude of neutral wind components. For the horizontal wind, we assume that the estimates follow a normal distribution of mean zero and uncertainty of 200 m/s. However, we expect that the vertical neutral wind

components are somewhat smaller, and decrease the uncertainty to 100 m/s. These constraints correspond to zeroth order Tikhonov regularization of the neutral wind with using 0.005 s/m and 0.01 s/m as the regularization parameter.

## 4.3    Boundary problems

With these statements, we can proceed with finding the uncertainty in estimated electric field and neutral wind. The different solutions to handle the boundary problems also impose some properties of the neutral wind and electric field estimates. We

did a short investigation of the different solutions as shown in Fig. 2. Except for ignoring all border-crossing non-symmetric derivatives, all solutions give results. The best of the solutions in terms of estimate accuracy is the symmetric derivative where we ignore those passing boundaries. When including them as stochastic variables the uncertainty is increased. This might be the most correct way of doing it, but further on we will ignore the boundary-passing derivatives because of simplicity, that is we are using the dark blue arrows in Fig. 2.

## 4.4    Accuracy of neutral wind and electric field estimates

The resulting uncertainties in the estimates of electric field for the coordinate system, measurements and regularization described in this section are shown in Figs. 5-7.  Like in the one-dimensional case investigated by Stamm et al. (2021a), the estimates of the electric field are somewhat accurate above 125 km altitude, while being quite uncertain below 125 km. According to the figures, estimates of the electric field is possible with a accuracy in the range of few millivolts per meter down

to 110-120 km altitude inside of the measured volume. Outside of the observed region, the electric field uncertainties grow. This is understandable since the measurements do not include information about the electric field at those locations. There, all information comes from the constraints.





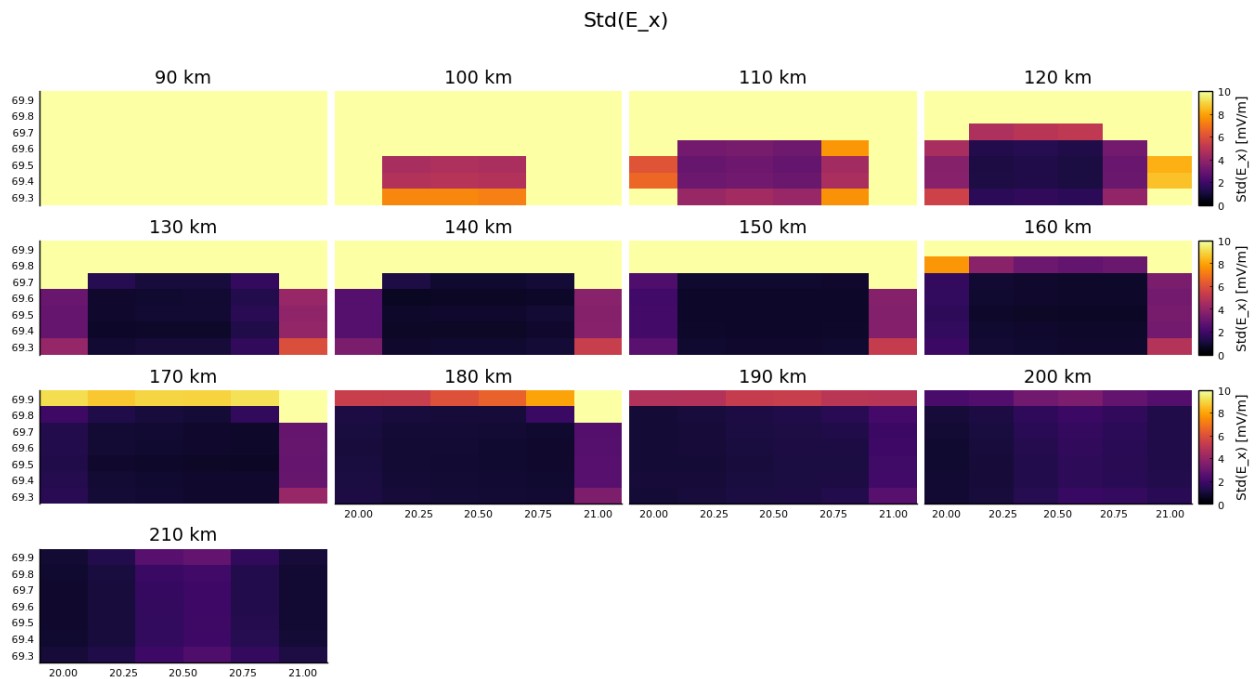

**Figure 5.** Uncertainty in electric field in local magnetic east direction.

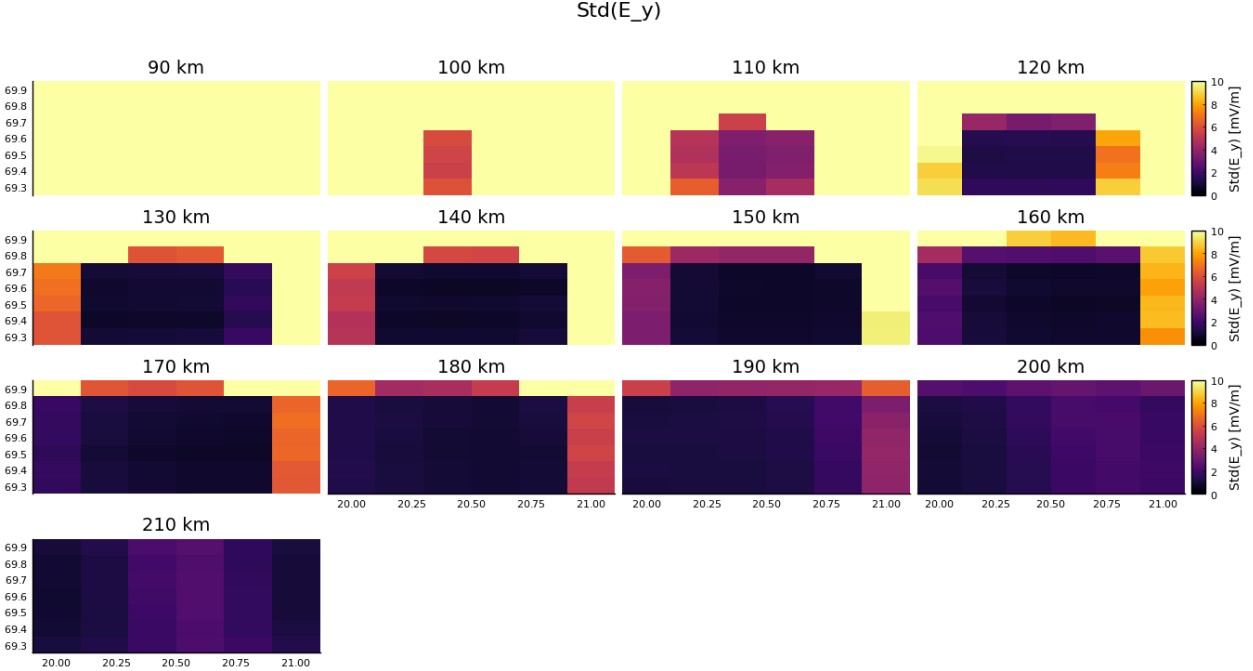

**Figure 6.** Uncertainty in electric field in local magnetic north direction.





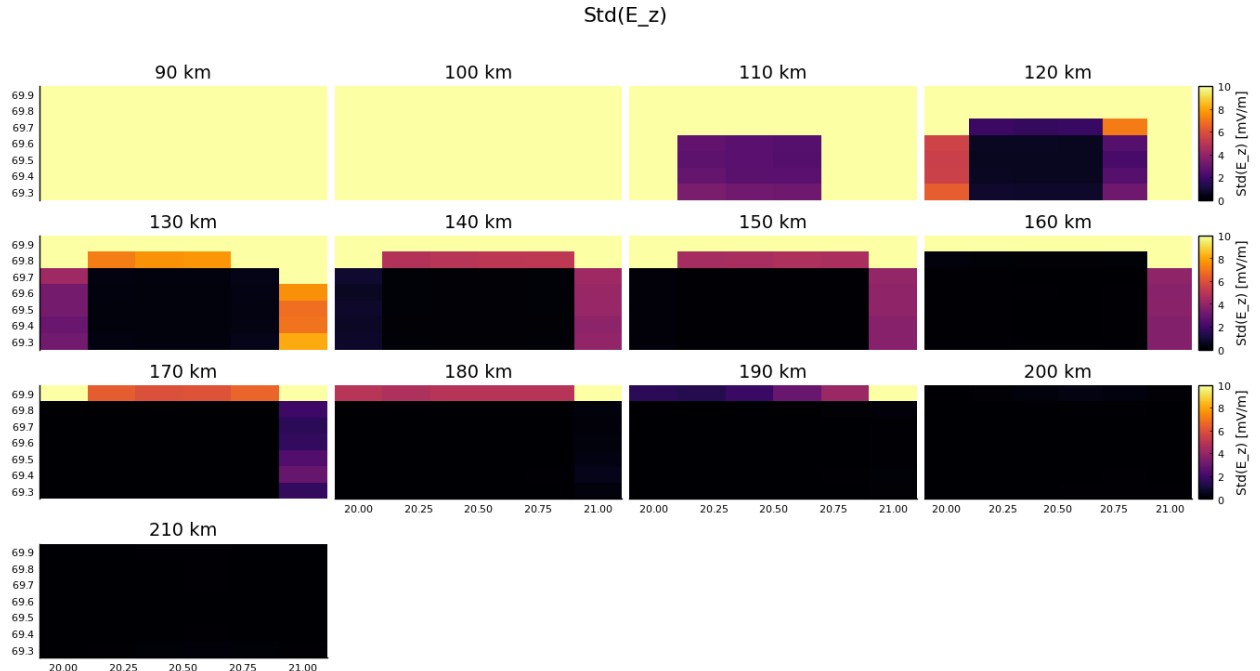

**Figure 7.** Uncertainty in electric field infield-aligned direction.

The uncertainties in neutral wind estimates are shown in Fig. 8. Also here, the same effect is observed, the neutral wind can be estimated with a high accuracy at low altitudes with a variance that increases rapidly above 110 km. The lowest estimates

for the neutral wind have accuracy of lower than 20 m/s below 120 km. These neutral wind estimates are slightly better than for the one-dimensional case. A reason could be our assumption on that the neutral wind has little variation horizontally because then, there are more measurements (beams) measuring the "same" neutral wind volume. As in the one-dimensional case, the accuracy of neutral wind measurements decreases with increasing altitude. It also seems to end at around the same value, namely 50 m/s.

**5  Simulation results**

In order to illustrate the results, we performed a simulation of vector field of neutral wind and electric field. We generated a vector field where the electric field in north-south direction points inward to a certain latitude, thereby simulating an auroral arc, similar to Nicolls et al. (2014). Inside of the arc, the field is zero. Also the other components of the electric field are set to zero. This can be compared to the Cowling channel model by Fujii et al. (2012). The neutral wind is set to zero everywhere.

We used the generated fields to simulate the ion velocities in the coordinate system example described in Sect. 4. Then, normally-distributed noise is added with standard deviation of 20 m/s in the horizontal directions and 5 m/s in the vertical





**Figure 8.** Uncertainty in neutral wind estimates. Because the uncertainties vary little horizontally, the values are averaged for every altitude.

direction. Finally, the simulated ion velocities are used to find estimates of neutral wind and electric field. Here, we use the same grids as for the generated fields and the regularizations as described in Sect. 4.1.

The generated vector fields for electric field and neutral wind are shown in Fig. 9 along with the ion wind measurements simulated from these. The estimated vector fields are shown in Fig. 10. The estimates where the uncertainty in at least one electric field component is above 10 mV/m are not plotted. Neither are those of neutral wind where at least one component has uncertainty above 30 m/s.

First of all, we note that the simulated ion velocity at the highest altitudes is perpendicular to the generated electric field. This is expected because at these altitudes, it is mainly influenced by the ExB-drift which was used by Brekke et al. (1973) to find electric field estimates. At lower altitudes, the ion drift becomes increasingly more dependent on the neutral wind.

The shown estimate of the electric field in Fig. 10 is quite close to the starting point at 125 km and upwards, but only inside of the measured volume. This is the same result as found in the one-dimensional case by Stamm et al. (2021a). We note that in the eastern boundary region of Fig. 10, there is a small curving artifact that is caused by Faraday's law.

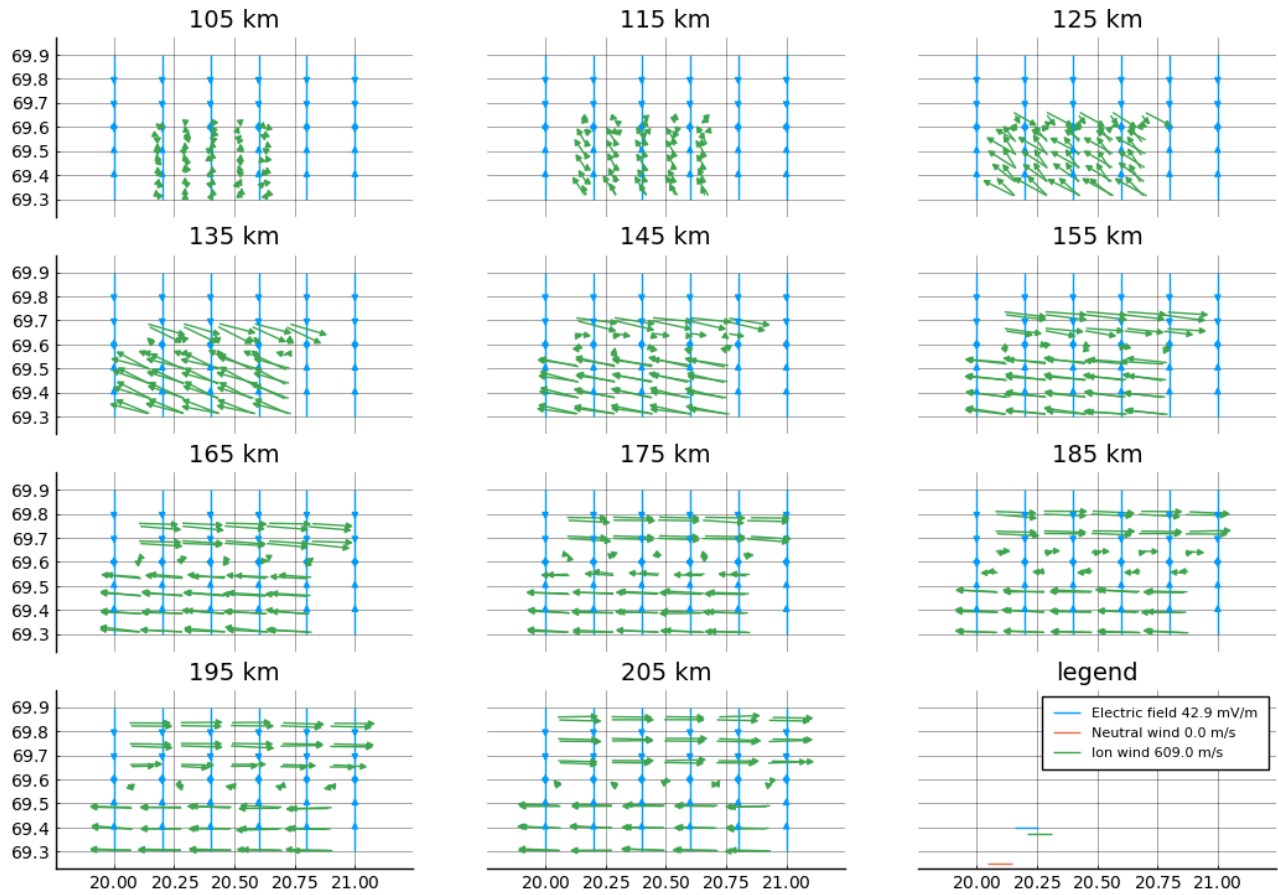

**Figure 9.** Electric field (blue) and neutral wind (red) used for simulations. Simulated ion wind measurements (green) are also shown. Because the neutral wind is set to zero it is not seen in the plot. The vertical spacing in the plot is chosen so that the first plot covers our model and measurements between 100 and 110 km range along the magnetic field, the second between 110 and 120 km and so on. Since there are measurements every fifth kilometer, each subplot contains two sets of measurements. For example, the 105 km plot contains the measurements from the line-of-sight ranges 100 km and 105 km. The plots for the uppermost and lowermost ranges look similar to their neighbour range and are not plotted.



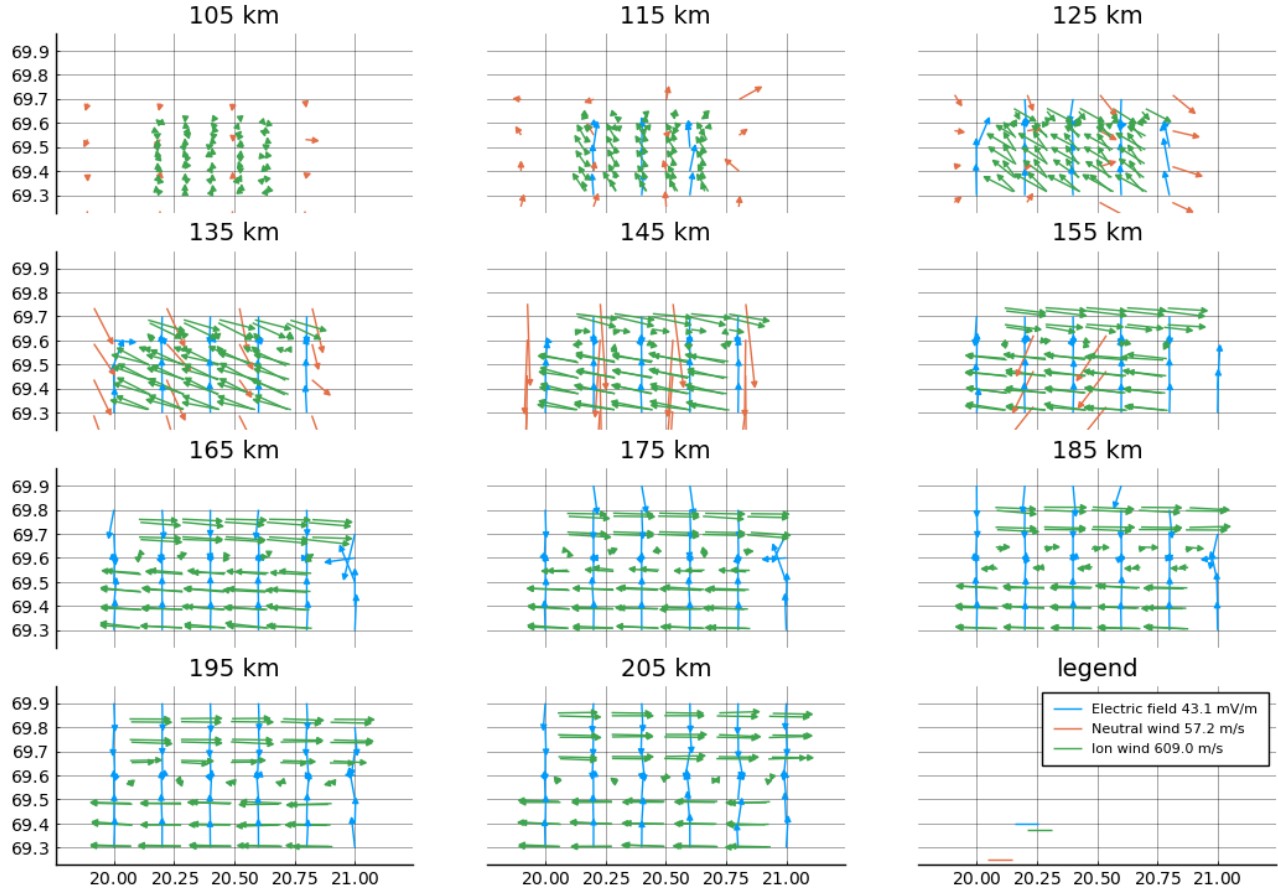

**Figure 10.** Estimated neutral wind (blue) and electric field (red) together with ion wind measurements (green). The plots for the uppermost and lowermost ranges look similar to their neighbour range and are not plotted. Also electric field vectors where at least one component has an uncertainty larger than 10 mV/m are not shown. Likewise, neutral wind vectors are not shown if one component has an uncertainty larger than 30 m/s.

Also, the neutral wind estimates can be described as somewhat correct below 125 km altitude. Those estimates above this become increasingly worse, like in the one-dimensional study.

# 6   Discussion and summary

This study introduces a method to estimate electric fields and neutral winds from multistatic multi-beam ISR measurements of ion velocity. We show that electric field uncertainties of few millivolts per meter can be achieved at altitudes above 120 km. Neutral wind estimate uncertainties should be small below 120 km. It is the extension into three dimensions which makes out the difference between this studies and Stamm et al. (2021a). The estimates from this three-dimensional technique give a more stable solution than in the one-dimensional case. Even if the study is more sophisticated in three dimensions, the approaches





give similar results which depend on the how the regularization is performed. In both cases, the results indicate that even with adding regularization, electric field and neutral wind cannot be estimated well at the same altitudes without further assumptions.

For the presented estimates from the simulated ion drifts, the advantage of using the previous neutral wind estimate is not used. By using the previous neutral wind estimates as a prior knowledge of the state of the neutral wind, the time-variation of the neutral wind estimates will be smoothed. This is similar to a Kalman-filtering approach. This approach allows us to take into account that the neutral wind changes slowly with time.

The inverse problem in this study involves a large scale of regularization parameters, and thereby parameters that can be adjusted. This results in some freedom in tuning the regularization parameters. When possible, we used weights for the

regularization terms that were taken from measurements of related parameters. Elsewhere, physical models or reasoning was used. However, it is possible that there are slightly better ways of constraining the problem or adjustments of the regularization parameters that, in some sense, give better results.

In the case of Faraday's law, we decided to increase the uncertainty of the time-derivative of the magnetic field from those values given by magnetometer data. We do this to allow for finer variations in the electric field estimates than else would be

allowed by our coarse grid.

As a performance test of the technique, we removed three of the central measurement beams, and estimated electric field and neutral wind from the remaining measurements. The estimates with measurements between 180 and 190 km altitude are shown in Fig. 11a, and the $E_\mathrm{x}$ uncertainties in Fig. 11b. The deviations relative to the estimates using the full set of measurements (cmp. with Fig. 10) are small. Maybe more importantly, the uncertainties do not increase by much. This shows that this type of

Tikhonov regularization leads to solutions that degrade gracefully while satisfying Maxwell's equations. A consequence of this is that it should be possible to use sparser beams to estimate electric field and neutral wind. This can be used to either improve the time resolution or to expand the observed volume. However, the removal of beams comes with a cost of slightly increased uncertainties, which can be seen by comparing Fig. 11b and Fig. 5.

*Code availability.* The code is currently only available on request from the corresponding author

*Author contributions.* JV came up of the idea and programmed programs for ISR spectrum and geographic calculations. JS programmed the model, carried out the calculations and prepared the article draft. AS, BG, JS and JV participated in developing the technique, the scientific discussions and the writing.

*Competing interests.* The authors declare that they have no competing interests

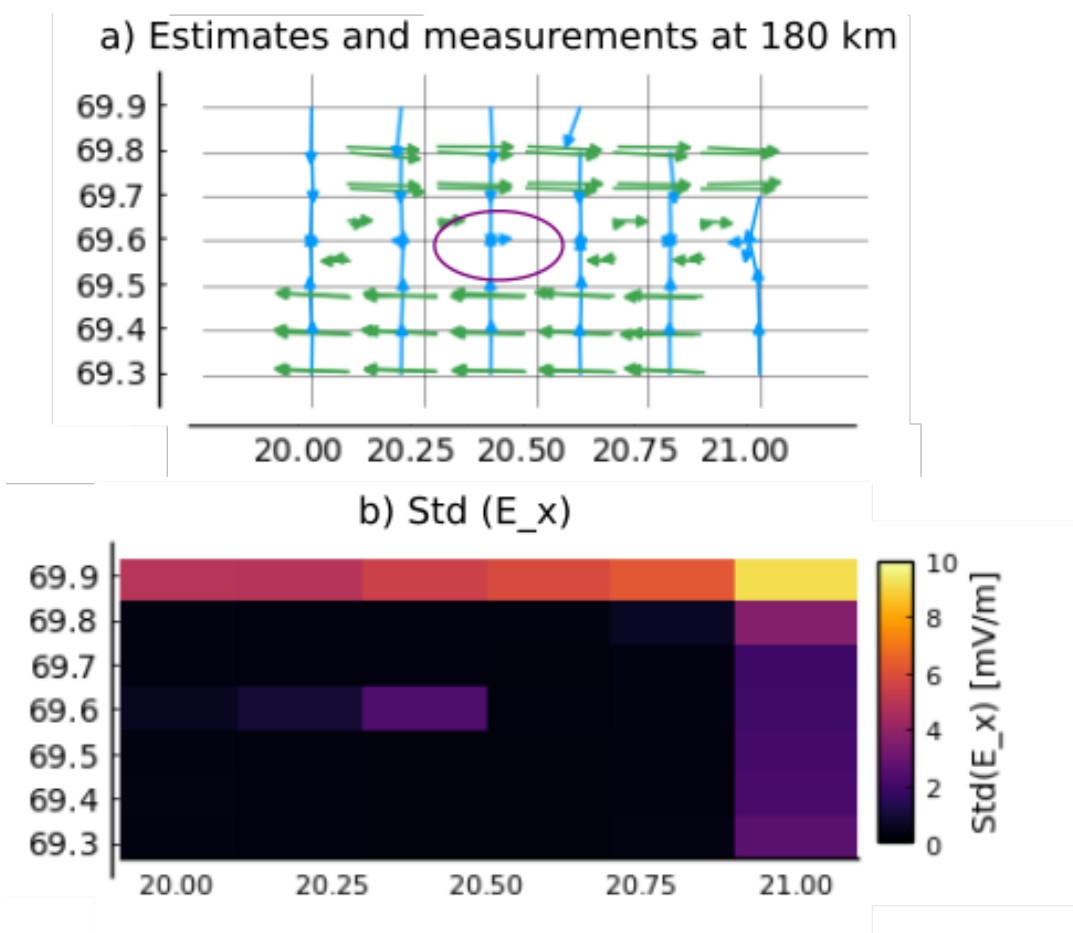

**Figure 11.** Estimates of electric field with measurement gap. Three central measurement beams have been removed. Figure a shows the remaining measurements and new estimates of electric field between 180 and 190 km. At other altitudes, the estimates show similar changes compared with Fig. 10. Figure b shows the corresponding uncertainties in "northward" electric field. At other altitudes, these show similar changes compared to Fig. 5.





**Acknowledgements**

This research has been supported by the Tromsø Science Foundation as part of the project "Radar Science with EISCAT3D" and the Research Council of Norway, grant 326039. The publication charges for this article have been funded by a grant from the publication fund of UiT The Arctic University of Norway. EISCAT is an international association supported by research organisations in China (CRIRP), Finland (SA), Japan (NIPR and ISEE), Norway (NFR), Sweden (VR), and the United Kingdom (UKRI)





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
