# Peer review of "A technique for volumetric incoherent scatter radar analysis"

_Annales Geophysicae, 2022_

## Referee Comment (RC2)

**Review of "A technique for volumetric incoherent scatter radar analysis" by Stamm et al.**

This manuscript describes an inverse technique for volumetric velocity reconstructions with direct relevance to the future EISCAT_3D radar. The problem in question is fundamentally ill-posed, namely solving for six unknown vector components (3 components of electric field and 3 components of neutral wind) given only measurements of 3 variables (the three components of ion velocity). As an ill-posed problem, the solution is highly reliant on a priori assumptions, and the final solution is only reasonable if all the a priori assumptions are reasonable. The justifications given for the a priori assumptions in this manuscript are inadequate and require additional examination and explanation, as described below. Furthermore, the formulation of the inverse problem contains conceptual flaws.

**Major Comments**

1. **The manuscript does not handle the F-region parallel ion velocities correctly.** The momentum equation in Eq. 3 is a vector equation, and it is approximately valid for the two perpendicular components. Nonetheless, this equation is not valid for the parallel component at F-region altitudes. Using that fact that $v_\parallel \times \mathbf{B} = 0$ and assuming that $E_\parallel$ is small, the parallel component of Eq. 3 reduces to $v_\parallel = u_\parallel$, implying that the parallel ion velocities are always equal to the parallel neutral velocities. This is generally not true in the F-region. A proper treatment of ion parallel velocity in the F-region requires the inclusion of gravity, ion pressure gradients, and ambipolar electric fields. The ion inertia terms can also become important during times of rapidly varying ion upflow.

   In principle, EISCAT_3D measurements could be used to volumetrically reconstruct all three components of the F-region ion velocities, including the spatial variations of the ion upflow velocities. The algorithm presented in this manuscript, however, would fail to do that. This manuscript is not solving for $v_\parallel$, but instead solving for $E_\parallel$ and $u_\parallel$ assuming the two quantities are related to $v_\parallel$ through an invalid parallel momentum equation.

   Figure 8 shows low uncertainties in the vertical neutral wind estimates extending all the way up to 200 km altitude. This is unreasonable since the ion velocities that the radar measures become collisionally decoupled from the neutral velocities at high altitudes, meaning the radar data cannot actually be giving meaningful information on neutral velocities at those altitudes. This unreasonable result is a direct consequence of the invalid parallel momentum equation.

2. **For vector basis functions, the weights should generally be arrays not scalars.** Eqs.

12 and 13 are vector equations of the form:

$$\mathbf{E} = \sum_j \eta_j \boldsymbol{\Phi}_j$$

$$E_x = \sum_j \eta_j \Phi_{xj}$$

$$E_y = \sum_j \eta_j \Phi_{yj}$$

$$E_z = \sum_j \eta_j \Phi_{zj}$$

This implies that all three components of the basis function get the same weight, which is an unusual restriction. To allow the three components to vary independently, the coefficients should be allowed to be different for the different vector components, i.e.

$$\mathbf{E} = \sum_{i=1}^{3} \sum_j \eta_{ij} \boldsymbol{\Phi}_j$$

$$E_x = \sum_j \eta_{1j} \Phi_{xj}$$

$$E_y = \sum_j \eta_{2j} \Phi_{yj}$$

$$E_z = \sum_j \eta_{3j} \Phi_{zj}$$

To be general, the three different components of $\eta$ should be treated as three separate unknowns.

3. **The manuscript does not assume equipotential field lines and does not explain the rationale for allowing large variations in electric fields along a field line.** Past E-region neutral wind estimation techniques such as *Thayer* [1998] and *Heinselman and Nicolls* [2008] have always asserted that electric fields are invariant along field lines such that F-region measurements of the electric fields can be mapped into the E-region. The mapping of F-region electric fields into the E-region is crucial for all of these past studies of E-region neutral winds using ISR; without that assumption the ion momentum equation is unsolveable in the E-region. Past sounding rocket studies have demonstrated the reality of field line mapping using payloads that can measure electric fields independently of ion velocity [*Sangalli et al.*, 2009].

   In this manuscript the a priori standard deviation of the electric field gradient is allowed to be 20 mV/m per 2.5 km in all three directions, including along the field lines. This is equivalent to asserting that field-aligned mapping of the electric fields does not function between the F- and E-regions; fields of 50 mV/m in the F-region at 300 km can change by more that 100% over the distance to the E-region at 100 km.

Ignoring field-aligned mapping of electric fields between the E- and F-region makes the problem of estimating E-region neutral winds substantially more difficult, and it is clearly leading to unreasonable results in the examples presented. Figure 10 shows the algorithm estimates non-zero neutral winds at 125-135 km altitude in a truth model simulation where the true neutral wind is zero. This behavior is pathological and unreasonable. The results assert that the variance of the estimated electric fields at low altitudes is nearly infinitely large, when in reality electric field mapping should guarantee that electric fields at low altitudes should nearly match the fields at high altitudes.

4. **The justifications for the allowed magnitudes of $\nabla \cdot \mathbf{E}$ are inadequately justified.** Line 220 assumes a deviation from charge neutrality that is $10^{-6}$ with no justification. This leads to an assumed variance of $\nabla \cdot \mathbf{E}$ of $10^{-3}$ V/m$^2$. This is actually a very large value.

   An alternative way to estimate a typical value for $\nabla \cdot \mathbf{E}$ would be to start from the height-integrated current continuity equation [*Clayton et al.*, 2021].

   $$J_{\parallel} = \Sigma_P \nabla_{\perp} \cdot \mathbf{E} + \nabla_{\perp} \Sigma_P \cdot \mathbf{E} - \nabla_{\perp} \Sigma_H \cdot \left( \mathbf{E} \times \hat{b} \right)$$

   Ignoring the conductance gradient terms, this is approximately

   $$\nabla_{\perp} \cdot \mathbf{E} \approx \frac{J_{\parallel}}{\Sigma_P}$$

   Using typical values of $J_{\parallel} = 5 \times 10^{-5}$ A/m$^2$ and $\Sigma_P = 5$ S gives $\nabla_{\perp} \cdot \mathbf{E} = 10^{-5}$ V/m$^2$, which is two orders of magnitude smaller than what this manuscript assumes. Note that using the height-integrated current continuity equation provides and estimate of $\nabla_{\perp} \cdot \mathbf{E} = \frac{\partial E_x}{\partial x} + \frac{\partial E_y}{\partial y}$, but if the field-aligned variation of $\mathbf{E}$ is small (i.e. $\frac{\partial E_z}{\partial z} \approx 0$), then $\nabla \cdot \mathbf{E} \approx \nabla_{\perp} \cdot \mathbf{E}$.

5. **The use of ground-based magnetometer data for constraining $\frac{\partial \mathbf{B}}{\partial t}$ is unjustified.** Ground-based magnetometers located at least 100 km below the current sources in the E-region are not necessarily going to capture realistic estimates of $\frac{\partial \mathbf{B}}{\partial t}$ in the F-region, particularly in cases where trapped Alfvén waves are bouncing around in the ionospheric Alfvén resonator. Observations from the rocket literature actually can justify large values of $\frac{\partial \mathbf{B}}{\partial t}$. For example, the observations from *Akbari et al.* [2022] describe standing Alfven waves with amplitudes of $\Delta E = \pm 40$ mV/m, $\Delta B = \pm 100$ nT, and frequencies of 0.25-0.5 Hz. In this case $\frac{\partial B}{\partial t} \approx 2\pi f |B| = 2\pi \times 0.5$ Hz $\times 100$ nT $= 314$ nT/s.

   A caveat with this analysis is the 70 second integration time is going to average over fluctuations associated with 0.5 Hz Alfven waves. Nonetheless, the integration is still not going to remove Ultra Low Frequency Pc5 waves (2-7 mHz), which can also have significant amplitudes (100s of nT on the ground, meaning they are even larger in the ionosphere).

6. **The assumptions about the relationship between electron density fluctuations and neutral density fluctuations is unjustified.** A more direct way to estimate neutral density variations is to look directly at lidar measurements of gravity waves. For example, *Vargas et al.* [2019] cite wave amplitudes ranging from 0.77 to 8.4% of the ambient sodium density, with an average of 2.7%. The assumed neutral density fluctuation of 50% at 100 km in the manuscript is unreasonably large.

7. **The use of a zeroth-order Tikhonov regularization is going to bias the neutral wind estimates low.** The assumed a priori variance is 200 m/s, but auroral neutral wind jets over 300 m/s have been observed, for example in the JETS rocket mission.

**Minor Comments**

1. The figure quality is generally low, with the text in the axis labels being highly pixelated.

2. An azimuth-elevation plot of the beam geometry would substantially clarify the beam geometry. Figures 3 and 4 have so many lines on them that the 3D geometry is hard to see.

3. Lines 209 and 210 should specify the interpulse period assumed for this experiment and specify how many independent estimates of the ACF/Spectra are obtained in 2 s of integration.

**References**

Akbari, H., R. Pfaff, J. Clemmons, H. Freudenreich, D. Rowland, and A. Streltsov, Resonant alfvén waves in the lower auroral ionosphere: Evidence for the nonlinear evolution of the ionospheric feedback instability, *Journal of Geophysical Research: Space Physics*, *127*(2), e2021JA029,854, doi:10.1029/2021JA029854, 2022.

Clayton, R., et al., Examining the auroral ionosphere in three dimensions using reconstructed 2d maps of auroral data to drive the 3d gemini model, *Journal of Geophysical Research: Space Physics*, *126*(11), e2021JA029,749, doi:10.1029/2021JA029749, 2021.

Heinselman, C. J., and M. J. Nicolls, A bayesian approach to electric field and e-region neutral wind estimation with the poker flat advanced modular incoherent scatter radar, *Radio Science*, *43*(5), doi:10.1029/2007RS003805, 2008.

Sangalli, L., D. J. Knudsen, M. F. Larsen, T. Zhan, R. F. Pfaff, and D. Rowland, Rocket-based measurements of ion velocity, neutral wind, and electric field in the collisional transition region of the auroral ionosphere, *Journal of Geophysical Research: Space Physics*, *114*(A4), doi:10.1029/2008JA013757, 2009.

Thayer, J. P., Height-resolved joule heating rates in the high-latitude e region and the influence of neutral winds, *Journal of Geophysical Research: Space Physics*, *103*(A1), 471–487, doi:10.1029/97JA02536, 1998.

Vargas, F., G. Yang, P. Batista, and D. Gobbi, Growth rate of gravity wave amplitudes observed in sodium lidar density profiles and nightglow image data, *Atmosphere*, *10*(12), doi:10.3390/atmos10120750, 2019.

---

## Author Response (AR1)

**Answer to referee 1**

Referee comment:

> *The paper is expanding on a technique published by Nicolls et al. (2014) by using physical constraints to measure the electric field from ion velocities measured from ISR. The technique seems reasonable and the authors show how the method will work through simulation examples.*

> *The authors deal with the affects of the sensor e.g. finite beam width and range extent of the measurments within equation 10. There can be a time component to this as the plasma moves through the FOV it can move to different resolution voxels. The closest cross beam voxels at 100 km along the north south track are about 5 km. This can be similar to a blurring of the data. Will this result in a major change in the algorithm? I think for the most part it will just be an adjustment to the forward model but it may not be neccesary as you're just measuring velocities and not intrinsic plasma parameters with this technique. Plus the physics based regularization might help mitigate this impact.*

Answer:

Thank you for the valuable feedback

The model in the article does not consider time variations in other ways than that the neutral wind is assumed to vary slowly. There could be cases where a feature in electric field or neutral wind moves with a velocity large enough to be inside several voxels within one loop of transmit beams. For example the auroral arc could move over the field of view over this time. The electric field changes would then appear blurred in the output data.

Including a time component that considers changes in the measured plasma within one integration time into the model would be possible. It could be implemented similar to the dependency on the volume.

Another way of handling rapid time-variations could be to shorten the integration time. The price to pay for avoiding motion-blurring this way is increased uncertainties. One could then apply a Kalman filter to the time domain. This is outside the scope of this study.

For Figure 11, we removed three beams completely to see how missing beams effect estimates and uncertainties. This could also be interpreted as an extreme valley in electron density. The result was that these are little affected. We expect that the technique could be applied in cases where faster scan times are used with beams in more sparse pointing directions leading to slightly larger uncertainties.

At the end of the discussion, we have added the following paragraphs:

«The presented framework assumes that the ionosphere does not change faster than the integration time, which is 70 s for the presented example. Spatial and temporal variations occurring faster that the integration time will thus be blurred out. One way to mitigate this is to take into account in which direction the beam points at every point in time such that the model connects the time the measurement is taken to the results. Another possible mitigation procedure, is to use a shorter integration time. The latter will have increased uncertainty which may be compensated to some extent by a Kalman filter. A third option would be to use fewer beams as this needs shorter integration time. The regularization will then try to fill the gaps as best as possible as illustrated in the example above.

In general we can state that an improved time-resolution requires measurements in fewer pointing-directions, that is either covering a smaller volume with a compact set of beams or a more sparse set of beams covering a large volume.»

**Answer to referee 2**

We thank the referee for the detailed review and comments. We will answer the comments in the order they appear.

We have included most of the referee comments before our answer such that it is easier to see what we are referring to. The referee comments are written in italics.

**Major comments**

1.

> *The manuscript does not handle the F-region parallel ion velocities correctly. The momentum equation in Eq. 3 is a vector equation, and it is approximately valid for the two perpendicular components. Nonetheless, this equation is not valid for the parallel component at F-region altitudes. Using that fact that $v_\parallel \times B = 0$ and assuming that $E_\parallel$ is small, the parallel component of Eq. 3 reduces to $v_\parallel = u_\parallel$ , implying that the parallel ion velocities are always equal to the parallel neutral velocities. This is generally not true in the F-region. A proper treatment of ion parallel velocity in the F-region requires the inclusion of gravity, ion pressure gradients, and ambipolar electric fields. The ion inertia terms can also become important during times of rapidly varying ion upflow.*

> *In principle, EISCAT 3D measurements could be used to volumetrically reconstruct all three components of the F-region ion velocities, including the spatial variations of the ion upflow velocities. The algorithm presented in this manuscript, however, would fail to do that. This manuscript is not solving for $v_\parallel$, but instead solving for $E_\parallel$ and $u_\parallel$ assuming the two quantities are related to $v_\parallel$ through an invalid parallel momentum equation.*

> *Figure 8 shows low uncertainties in the vertical neutral wind estimates extending all the way up to 200 km altitude. This is unreasonable since the ion velocities that the radar measures become collisionally decoupled from the neutral velocities at high altitudes, meaning the radar data cannot actually be giving meaningful information on neutral velocities at those altitudes. This unreasonable result is a direct consequence of the invalid parallel momentum equation.*

Answer:

This study is for showing that it is possible to use multi-beam multi-static ISR measurements to obtain volumetric measurements of electric field and neutral wind in the ionosphere. In that sense, we did some simplifications of the momentum equation where we neglected the smallest terms, that is advection, gravity and pressure gradients. It is possible to extend the model to include these terms. This will make the model more correct, but also requires even more calculations.

The shown uncertainties do not include those introduced by assumptions or simplifications made. The neglected influence of advection, gravity and pressure gradient terms on the velocities are therefore not reflected as increased uncertainties. Typically, these terms are small, but in the

uppermost part of our volume, they can be large enough to become significant. If future work uses field-aligned/vertical estimates of neutral wind or electric field, they will have to consider these terms.

In the revised manuscript, we will state the momentum equation including the terms mentioned above explicitly, before we simplify it to obtain the equation we use in the analysis. We will also clarify that the assumptions and simplifications we did are not included in the shown uncertainties.

In the revised manuscript we have included the momentum equation including the neglected terms. We also explain the steps and approximations between this equation and the momentum equation used.

2.

> For vector basis functions, the weights should generally be arrays not scalars. [...][It is implied that] all three components of the basis function get the same weight, which is an unusual restriction. To allow the three components to vary independently, the coefficients should be allowed to be different for the different vector components [...]

> To be general, the three different components of η should be treated as three separate unknowns.

Answer:

It appears that the text was confusing since the the three components of the basis functions are allowed to vary independently in the calculations and simulations.

This will be clarified in the revised manuscript.

Changes:

In the revised manuscript, we have tried to clarify that the components are independent by adding subscripts to the equations 13 and 14 indicating the component.

3.

> The manuscript does not assume equipotential field lines and does not explain the rationale for allowing large variations in electric fields along a field line. Past E-region neutral wind estimation techniques such as Thayer [1998] and Heinselman and Nicolls [2008] have always asserted that electric fields are invariant along field lines such that F-region measurements of the electric fields can be mapped into the E-region. The mapping of F-region electric fields into the E-region is crucial for all of these past studies of E-region neutral winds using ISR; without that assumption the ion momentum equation is unsolveable in the E-region. Past sounding rocket studies have demonstrated the reality of field line mapping using payloads that can measure electric fields independently of ion velocity [Sangalli et al., 2009].

> In this manuscript the a priori standard deviation of the electric field gradient is allowed to be 20 mV/m per 2.5 km in all three directions, including along the field lines. This is equivalent to asserting that field-aligned mapping of the electric fields does not function between the F- and E-regions; fields of 50 mV/m in the F-region at 300 km can change by more that 100% over the distance to the E-region at 100 km.

> *Ignoring field-aligned mapping of electric fields between the E- and F-region makes the problem of estimating E-region neutral winds substantially more difficult, and it is clearly leading to unreasonable results in the examples presented. Figure 10 shows the algorithm estimates non-zero neutral winds at 125-135 km altitude in a truth model simulation where the true neutral wind is zero. This behavior is pathological and unreasonable. The results assert that the variance of the estimated electric fields at low altitudes is nearly infinitely large, when in reality electric field mapping should guarantee that electric fields at low altitudes should nearly match the fields at high altitudes.*

Answer:

It is accurately noticed that we do not assume that the electric potential maps perfectly along the magnetic field. This allows our method to be used also in cases where this assumption can not be done.

The mapping of electric field along the magnetic field lines is not perfect. This is especially true on smaller scale structures or in the lower E region, as shown by Reid (1965) and Park and Dejnakarintra (1974) (see also Brekke (2013)). This is also seen in Sangalli et al. (2009) where the electric field in the E region fluctuates.

The allowed variances for the electric field gradients is not considered in itself, but through Maxwells equations. However, there are no neighbouring voxels at the borders, which means that these gradients can not be implemeted directly with differences between the neighbours. We thought about several techniques to handle this issue, which are shown in Figure 2 of the paper. Three of the techniques to handle the borders give border-crossing derivatives a larger uncertainty. This uncertainty is 20 mV/m per 2.5 km (1 m/s per km for neutral wind).

Also, as described in Section 4.3, the setup we use for the borders give the same uncertainty to gradients over the border as to those within the borders.

Apparently, our manuscript could be understood as we would use these constraints throughout the volume, possibly in all directions. We will try to clarify the manuscript.

Changes:

We have done some formulation changes in the corresponding paragraph trying to clear up

4.-7.

(We did not include the referee comments here because of the large content of equations)

Answer:

We gratefully thank the referee for suggesting more reasonable strictnesses of the constraints we use. In general, those are stricter than those we use in our calculations. We performed new simulations using the constraints suggested by the referee in points 4-7 and letting the zeroth order Tikhonov regularization have an a priori standard deviation of 300 m/s. As a result, the electric field remains unchanged, but the neutral wind estimates become more varying and uncertain. To compare, we also tested how the model would react to the same situation if the wind blows with 300 m/s. Also here, the different strictnesses of regularization give small differences.

In general, one should not regularize an inverse problem more than necessary. Therefore the least strict assumptions are preferable because they then give more room for unforeseen variations in the

ionosphere. We therefore do not want to tighten the constraints since it does not seem necessary. We will include these reasonings and references into the manuscript.

Changes:

We have extended the discussion or reasoning for every of these comments on regularization parameters in the revised manuscript. These are placed where the regularization parameters are described, except in the case of Gauss' law where it is placed in the discussion section.

We have let most of the regularization parameters unchanged as the those proposed by the referee are in the same order of magnitude as the used ones. See also additional changes below.

We have added a paragraph in the middle of section 3 trying to describe the purpose of the regularization better.

7.

*> The use of a zeroth-order Tikhonov regularization is going to bias the neutral wind estimates low. The assumed a priori variance is 200 m/s, but auroral neutral wind jets over 300 m/s have been observed, for example in the JETS rocket mission.*

Answer:

It is correct that zeroth-order Tikhonov regularization biases the solution towards the a priori expected value, here zero. However, this bias is not as strong as the referee seems to suggest. The assumed standard deviation is 200 m/s - this means that winds of 300 m/s still would be at 1.5-sigma. A standard deviation for the wind components of 200 m/s would lead to a small under-estimate of these components of a few tens of metres per second.

**Minor comments**

1.

*> The figure quality is generally low, with the text in the axis labels being highly pixelated.*

We will increase the figure resolution in the next version of the manuscript.

Changes:

The figures showing results are now in pdf format and with larger labels.

2.

*> An azimuth-elevation plot of the beam geometry would substantially clarify the beam geometry. Figures 3 and 4 have so many lines on them that the 3D geometry is hard to see.*

We will include a plot showing the azimuth-elevation distribution of transmit beams.

Changes:

The desired plot is now included as figure 5. We added the direction of the magnetic field at ground into that figure.

3.

> *Lines 209 and 210 should specify the interpulse period assumed for this experiment and specify how many independent estimates of the ACF/Spectra are obtained in 2 s of integration.*

Changes:

We have include this information to the manuscript.

**References**

Brekke, Asgeir (2013): Physics of the upper polar atmosphere. 2. edistion, Springer, Heidelberg.

Park, C. G. and Dejnakarintra, M. (1974): Paper presented at Fifth International Conference on Atmospheric Electricity, Garmisch-Partenkirchen, West Germany, September 2–7, 1974.

Reid, G. C. (1965): Ionospheric effects of electrostatic fields generated in the outer magnetosphere. Radio Science, 69D, pp. 827–837.

Sangalli, L., D. J. Knudsen, M. F. Larsen, T. Zhan, R. F. Pfaff, and D. Rowland (2009): Rocket-based measurements of ion velocity, neutral wind, and electric field in the collisional transition region of the auroral ionosphere, Journal of Geophysical Research: Space Physics, vol. 114 number A4, doi: 10.1029/2008JA013757.

**Additional changes**

After having written the author comments, we detected that we had used a wrong mass density of air for calculating the changes in the neutral density. A more correct value can for example be taken from MSIS, $10^{-7}$ kg/m³. We decided to apply the reasoning from referee 2 to find a parameter for constraining with the continuity equation.

After this change, the continuity equation constrains the solutions little, meaning that results are not changed importantly.

We have also changed dr to dV in the integrals in equations 15 and 16 to make them comply better with the integrals before.